# Early Age- and Sex-Dependent Regulation of Astrocyte-Mediated Glutamatergic Synapse Elimination in the Rat Prefrontal Cortex: Establishing an Organotypic Brain Slice Culture Investigating Tool

**DOI:** 10.3390/cells12232761

**Published:** 2023-12-04

**Authors:** Eugenia Vivi, Lea R. Seeholzer, Anastasiia Nagumanova, Barbara Di Benedetto

**Affiliations:** 1Laboratory of Neuro-Glia Pharmacology, Department of Psychiatry and Psychotherapy, University of Regensburg, 93053 Regensburg, Germany; eugenia.vivi@ukr.de (E.V.); lea.seeholzer@tum.de (L.R.S.); anastasiia.nagumanova@stud.uni-regensburg.de (A.N.); 2Regensburg Center of Neuroscience, University of Regensburg, 93053 Regensburg, Germany

**Keywords:** organotypic brain slice culture, astrocyte, synaptic phagocytosis, critical period, sex differences

## Abstract

Clinical and pre-clinical studies of neuropsychiatric (NP) disorders show altered astrocyte properties and synaptic networks. These are refined during early postnatal developmental (PND) stages. Thus, investigating early brain maturational trajectories is essential to understand NP disorders. However, animal experiments are highly time-/resource-consuming, thereby calling for alternative methodological approaches. The function of MEGF10 in astrocyte-mediated synapse elimination (pruning) is crucial to refine neuronal networks during development and adulthood. To investigate the impact of MEGF10 during PND in the rat prefrontal cortex (PFC) and its putative role in brain disorders, we established and validated an organotypic brain slice culture (OBSC) system. Using Western blot, we characterized the expression of MEGF10 and the synaptic markers synaptophysin and PSD95 in the cortex of developing pups. We then combined immunofluorescent-immunohistochemistry with Imaris-supported 3D analysis to compare age- and sex-dependent astrocyte-mediated pruning within the PFC in pups and OBSCs. We thereby validated this system to investigate age-dependent astrocyte-mediated changes in pruning during PND. However, further optimizations are required to use OBSCs for revealing sex-dependent differences. In conclusion, OBSCs offer a valid alternative to study physiological astrocyte-mediated synaptic remodeling during PND and might be exploited to investigate the pathomechanisms of brain disorders with aberrant synaptic development.

## 1. Introduction

Astrocytes represent the most abundant subtype of glial cells in the central nervous system. They are characterized by a typical star-shaped morphology with numerous ramified processes and exert a wide range of distinct functions, e.g., the support of endothelial cells at the blood–brain barrier, the maintenance of ion homeostasis around synapses and in the extracellular space, the metabolic support of neuronal survival and the regulation of synaptogenesis, among others [1,2,3,4,5]. Several clinical studies of neuropsychiatric (NP) disorders show altered astrocyte properties [6,7,8,9,10,11] and pre-clinical studies have been pivotal in clarifying the neurobiological underpinnings of those disorders [12,13,14,15].

In rodents, the generation, expansion and maturation of astrocytes (astrogenesis) take place during the first three weeks of brain postnatal development (PND) and are for the most part completed by the end of the so-called “critical period”, a highly sensitive time window of elevated brain plasticity [16,17,18]. In humans, this developmental time frame corresponds to the juvenile period of postnatal brain growth, which ends around the first decade of life, when adolescence begins [19]. However, it is still debated whether it might further extend into adolescence and even early adulthood. This latter aspect becomes clearly relevant when investigating putatively detrimental effects of environmental cues on early brain developmental processes or searching for the most effective therapeutic treatments for early-onset mental disorders, thereby calling for further investigations [20,21].

Astrogenesis, together with the refinement and specialization of astrocyte processes [22], coincides with a highly active period of synaptogenesis, which culminates with the establishment of properly functional neuronal networks [23,24]. The work of Chung and colleagues revealed that in mice, astrocytes play a pivotal role in synapse elimination (pruning), mediated by the multiple EGF-like domains 10 (MEGF10) phagocytic pathways. This process is prevalently directed toward glutamatergic synapses and occurs during both developmental stages and adulthood. MEGF10, the mammalian ortholog of the glia-specific phagocytic proteins Draper in *Drosophila melanogaster* and CED-1 in *Caenorhabditis elegans*, is a transmembrane receptor almost exclusively expressed in astrocytes [25,26].

MEGF10 can mediate its phagocytic functions on apoptotic material via the recognition of the “eat-me signal” C1q bound to phosphatidylserine (PS) exposed on dying cells [27]. Interestingly, Chung and colleagues showed that the MEGF10 phagocytic pathway is essential for the establishment of the eye-specific retinogeniculate segregation and the maintenance of functional synaptic homeostasis in the adult hippocampus [26,28]. Any disturbance in the sequence of these events may lead to the development of dysfunctional neuronal circuits and improper synaptic transmission, thereby contributing to the onset of brain disorders [29].

Among additional factors that may affect brain development, sex differences have been consistently reported in epidemiological studies to be relevant due to their impact on both physiological and pathological processes. However, there is a gap in research when it comes to unravelling the causes of these differences, thereby preventing not only a deep understanding of sex-dependent neurobiological underpinnings of synaptic network development, but also the etiological and pathogenic mechanisms of mental disorders in general. Moreover, this lack of knowledge hinders the generation of therapeutic approaches specific for either women or men [30,31].

The prefrontal cortex (PFC) is one of the regions with high vulnerability to environmental and endogenous stimuli, especially during sensitive early PND stages. Exposure to detrimental cues during these early growing phases has the potential to leave lasting imprints on the developing system, ultimately shaping maladaptive adult behaviors [19,32]. Previous studies in humans have carefully described changes in synaptogenesis from childhood through adolescence into early adulthood [33]. In rats, sex-dependent differences have been observed in the size of the PFC in adulthood, which started to appear during adolescence and could stem from neuronal death or pruning effects on neuronal circuits [34].

Primary dissociated cultures have been pivotal to study molecular, morphological and biochemical features of single-homogeneous-cell populations, allowing for the performance of animal research experiments, without excessively increasing the numbers of experimental animals or their sufferance. However, in vitro cell cultures inherently lack 3D structures to explore the functional interplays between brain cells in a more complex context, where the main cell–cell physical and mechanical interactions are preserved. Thus, several models have been developed to support the analysis of such interactions. Among them, organotypic brain slice cultures (OBSCs) have proven to be useful for investigating specific cellular and molecular brain processes ex vivo [35,36,37]. Moreover, they maintain various aspects of the structural and synaptic organization of the original tissue. With respect to other in vitro/ex vivo systems, such as the more complex brain organoids [38,39], OBSCs represent a useful tool with various advantages, notwithstanding some evident limitations [40,41].

Here, we first utilized Western blot to measure the expression of MEGF10 and the glutamatergic synaptic markers synaptophysin (presynaptic) and postsynaptic density-95 (PSD-95; postsynaptic) during the critical period of brain development, at postnatal (P) day 7, P14 and P21, and after its closure, at P32, in the cortex of male and female Wistar rats. In parallel, we used immunofluorescent immunohistochemistry combined with the 3D reconstruction of single astrocytes to examine the age-dependent and sex-specific differences in astrocyte-mediated synapse elimination more closely into the infralimbic/prelimbic areas of PFC. Furthermore, we established and validated an OBSC model, which might serve to evaluate how the MEGF10 phagocytic pathway contributes to the formation of glutamatergic synaptic networks in the PFC during the critical period and whether it might be implicated in brain disorders with synaptic aberrancies. The application of pharmacological and/or genetic manipulations in OBSCs may be helpful in identifying and investigating novel, sex-specific, disease trajectories and to potentially develop alternative treatment options tailored to the needs of males and females.

## 2. Materials and Methods

### 2.1. Animals

Experiments were carried out using male and female Wistar rats at postnatal days 0 to 32. Animals were group-housed under standard conditions (12 h light/dark cycle: lights on at 06:00 a.m., 22–24 °C, 55% humidity, ad libitum access to food and water). Pregnant dams were acquired from Charles River Laboratories (Charles River, Sulzfeld, Germany). All experimental procedures were approved by the government of Oberpfalz, Germany, and performed in accordance with the guidelines of the European Community Council Directives of 1 January 2013 (2010/63/EU). All efforts were taken to minimize animal pain or discomfort.

### 2.2. Developing Pups

#### 2.2.1. Protein Isolation and Immunoblotting

Whenever possible, mostly prefrontal cortex (PFC) tissue was collected at different postnatal stages (P0, P7, P14, P21, P32) and incubated in an ice-cold RIPA buffer containing 50 mM TRIS-HCl, pH 7.5 and a protease inhibitor cocktail (Roche) for 15 min. However, the tissue extracted might have included proteins from neighboring areas (Figure 1A). Subsequently, homogenates were sonicated (4 × 20 s, 20% intensity) and centrifuged at 15,000 rcf for 25 min at 4 °C. The supernatant containing proteins was collected and stored at −20 °C.

The samples were boiled for 5 min at 95 °C and loaded onto 10% or 12% SDS-PAGE, depending on the size of the protein of interest (10% for MEGF10, 12% for PSD95 and synaptophysin). A total amount of 12 µg of protein per lane was loaded to be separated via electrophoresis and transferred to nitrocellulose membranes (Protran BA, Whatman, GE Healthcare, Munich, Germany). However, some technical variations might still occur during the loading procedure, possibly affecting the evenness of the total proteins run and transferred. To address this, we consistently included the analysis of internal loading controls, such as cofilin or β-actin, to correct for such putative variations and allow for relative comparisons of results. Membranes were blocked with 5% milk or BSA in Tris-buffered saline + Tween-20 (TBST) and incubated overnight at 4 °C with primary antibodies as follows: anti-MEGF10 antibody (1:500, Thermofisher, Cat#PA5-76556, Waltham, MA, USA), anti-PSD95 antibody (1:2500, Sigma-Aldrich, Cat#MAB1596, St. Louis, MO, USA), anti-synaptophysin antibody (1:5000, Abcam, Cat#ab52636, Cambridge, UK), anti-β-actin antibody (1:10,000, Abcam, Cat#ab179467) and anti-cofilin antibody (1:1000, Cell Signaling, Cat#5175S, Danvers, MA, USA). Membranes were washed and incubated with horseradish peroxidase (HRP)-conjugated anti-mouse (1: 2500, Dianova, Cat#115-035-003, Hamburg, Germany) and anti-rabbit (1:1000 and 1:5000 for β-actin, Dianova, Cat#111-035-003) IgG antibody for 2 h. Immunoreactivity was detected using the enhanced chemiluminescence kit (ECL, SuperSignal chemiluminescent substrate, Thermofisher, Cat#34580). The intensity of the bands was scanned and analyzed quantitatively using ImageJ software (http://imageJ.nih.gov/ij, Java 1.8.0_172). The immunoreactivity of the proteins of interest was normalized to that of either β-actin or cofilin, as specified in the respective figure legends. In all gels, a sample was loaded to guarantee comparability of results, even when samples had to be run in different gels because of space limitations. Specifically, we used P0/P7 and DIV7 samples for pups and OBSCs, respectively. For the preparation of images, representative selected bands of the results were cropped from original gels and put together in a single panel to enhance their readability. In a few cases, the brightness/contrast tool was applied, but only at the end, to the whole picture. The full uncropped gels from which the bands were taken are collated in Appendix A.

#### 2.2.2. Immunofluorescent Immunohistochemistry to Examine Synaptogenesis and Astrocyte Phagocytosis in Developing Pup Brains

For immunostaining, male and female littermates were either anesthetized with CO_2_ or directly decapitated. Intact hemibrains were collected in ice-cold phosphate-buffered saline (PBS) and fixed in 4% paraformaldehyde/PBS for two weeks at 4 °C followed by cryoprotection in 30% sucrose. Afterward, hemibrains were sectioned coronally at 40 µm and preserved in 25% ethylene glycol/25% glycerol in PBS at −20 °C. Brain sections were then washed thoroughly before permeabilization and blocking in 0.2% Triton-X100+2% Normal Goat Serum (NGS) in PBS for 1.5 h. Brain slices were incubated overnight at 4 °C with primary antibody solution as follows: anti-LAMP1 antibody (1:100, Abcam, Cat#ab24170), anti-synaptophysin antibody (1:500, Synaptic System, Cat#101 006, Göttingen, Germany), anti-GFAP antibody (1:250, Sigma-Aldrich, Cat#G3893), anti-S100β antibody (1:500, Abcam, Cat#ab11178) in 0.2% Triton-X100+2% NGS in PBS. After washing, sections were incubated with secondary antibodies Alexa Fluor 488-conjugated anti-chicken IgG (1:500, Invitrogen, Cat#A11039, Waltham, MA, USA), Cy3-conjugated anti-mouse IgG (1:400, Sigma-Aldrich, Cat#C2181) and biotin-conjugated anti-rabbit IgG (1:500, Dianova, Cat#111-065-003) diluted in 2% NGS in PBS for 1.5 h. After washing, a solution containing DAPI (1:1000, Merck, Cat#32670, Darmstadt, Germany) and Alexa Fluor 647-conjugated Streptavidin (1:1000, Thermofisher, Cat#S21374) diluted in 2% NGS in PBS was applied to samples for 1 h. Finally, the brain sections were mounted on slides for confocal analysis.

#### 2.2.3. Confocal Imaging and Analysis

Confocal images from the infralimbic/prelimbic areas of the PFC were taken using the Olympus FV31S-SW confocal microscope (Olympus Europe Holding GmbH, Hamburg, Germany). The images were acquired from at least two slices per animal (n = 4 animals/time point) with a 60× magnification plus 2× optical zoom. Twenty-five z-stack images of 512 × 512 µm were taken at 0.50 µm intervals. To examine the overall rate of synaptogenesis locally, the total synaptophysin fluorescence intensity per field was calculated using corrected total cell fluorescence (CTFC) on ImageJ, after subtracting the background noise. To analyze astrocyte internalization of synaptophysin in lysosomes (LAMP1 positive (LAMP1+) puncta), images were reconstructed using IMARIS 9.8 software (Bitplane, Zurich, Switzerland). In brief, single astrocytes were 3D-reconstructed using the surface rendering function. Both LAMP1 and synaptophysin channels were masked inside the astrocyte surface and synaptophysin puncta inside the lysosomes were quantified using the ‘spot’ function. Astrocyte phagocytosis was assessed using the IMARIS 9.8 software (Bitplane, Zurich, Switzerland). In brief, individual astrocytes were selected and cropped (‘crop tool’) and subjected to 3D reconstruction utilizing the ‘surface rendering’ function. Subsequently, the LAMP1 and synaptophysin channels were specifically masked within the identified astrocytic surface. Quantification of LAMP1+ spots was executed utilizing the ‘spot’ function, while synaptophysin+ spots were redefined within the LAMP1+ spots, with exclusive quantification restricted to those residing within lysosomes. The phagocytic index was calculated by determining the total number of synaptophysin+/LAMP1+ colocalizing spots normalized to the total volume of a given astrocyte. This normalization to the volume was used to equalize putative differences among astrocytes due to the staining procedure with antibodies, which might not have always uniformly labelled the entire astrocyte. Moreover, the same number of astrocytes (six) per animal was analyzed at any given developmental stage to enable the comparison of age-dependent changes in the phagocytic index, independently from developmentally driven increases in total numbers of astrocytes along the PND stages.

### 2.3. Organotypic Brain Slice Cultures (OBSCs)

#### 2.3.1. Organotypic Brain Slice Preparation

Organotypic brain slices (OBSCs) were prepared employing the method of Stoppini et al. [35] and Humpel [42]. In brief, male and female rat pups aged postnatal day 4 (P4) to P6 were promptly decapitated in an ice-cold, oxygenated (95%/5% O_2_/CO_2_) dissection medium mixed with D-glucose (10 mM) and penicillin–streptomycin (100 U/mL), and the olfactory bulb and cerebellum were removed. Brains were mounted on the cutting disk with a thin layer of superglue. Sequential coronal slices containing the PFC (400 µm thickness, 9 slices of both hemispheres) were obtained using a vibratome platform (Leica VT1200, Leica Biosystems, Wetzlar, Germany) submerged in chilled oxygenated dissection solution at a speed of 0.06 mm/s. Subsequently, brain slices were carefully transferred onto translucent porous membranes (Millicell Cell Culture inserts, 0.4 µm, 30 mm, Merck Millipore, Cat#PICM03050, Burlington, MA, USA) pre-conditioned with 1 mL of the OBSC culture medium placed in 6-well plates.

#### 2.3.2. Organotypic Brain Slice Culture

OBSCs were cultured in groups of 3 slices per animal onto membrane inserts in a 1 mL OBSCs culture medium at 37 °C, 5% CO_2_ in 6-well plates. The OBSC culture medium consisted of the basal medium eagle with Earle’s balanced salt solution (EBSS), 25 mM HEPES, 25% inactivated horse Serum, 5 mg/mL D-glucose, 1 mM GlutaMAX and 1% penicillin–streptomycin. Half of the OBSC culture medium was replaced every 3 days (500 µL removed, replaced with 800 µL due to evaporation). OBSCs were kept in culture up to 21 days in vitro (DIV).

#### 2.3.3. Cell Viability Assay

The cell viability of OBSCs was assessed using a propidium iodide (PI) dye (Sigma-Aldrich, Cat#P4170). In brief, brain slices collected at DIV0, 7, 14 and 21 were placed in a fresh culture medium containing 2 µg/mL of PI and incubated for 30 min at 37 °C. Subsequently, OBSCs were transferred into new wells containing DAPI and incubated another 30 min at RT. Finally, OBSCs were mounted on slides and imaged using the inverted Zeiss LSM 880 Airyscan microscope with a 10× magnification. This method has been widely used to assess cell viability in OBSCs [43].

#### 2.3.4. Protein Isolation and Immunoblotting

Proteins were isolated from OBSCs at different time points: DIV7, 14 and 21. In brief, brain slices were washed with ice-cold PBS, cut out of the membrane inserts, and afterward, the cortices were dissected and collected in an ice-cold RIPA buffer for 15 min. For OBSCs, it was not really possible to selectively isolate PFC from other cortical regions, therefore the tissue analyzed may contain proteins from neighboring areas. Subsequently, tissue homogenates were sonicated (2 × 20 s, 20% intensity) and centrifuged at 15,000 rcf for 25 min at 4 °C. For immunoblotting and protein analysis, the same protocol was applied as described in Section 2.2.1.

#### 2.3.5. Immunofluorescent Immunohistochemistry to Examine Astrocyte Phagocytosis in OBSCs

Membranes containing brain slices were transferred into a 6-well plate and rinsed with PBS for 10 min. Slices were then fixed with gentle shaking in 4% paraformaldehyde in PBS at 4 °C overnight and washed three times with PBS at room temperature. Finally, the culturing membranes containing brain slices were cut out of the insert and placed in 0.1% sodium azide–PBS to be stored at 4 °C.

Following 2 h blocking and permeabilization at room temperature in 0.1% Triton-X100 + 2% NGS in PBS, fixed brain slices were labeled with anti-GFAP antibody (1:400, Sigma-Aldrich, Cat#G3893), anti-LAMP1 antibody (1:100, Abcam, Cat#ab24170) and anti-synaptophysin antibody (1:500, Synaptic System, Cat#101 006) in 0.1% Triton-X100+2% NGS in PBS, overnight at 4 °C. Slices were incubated with secondary antibodies Alexa Fluor 488-conjugated anti-chicken IgG (1:500, Invitrogen, Cat#A11039), Cy3-conjugated anti-mouse IgG (1:400, Sigma-Aldrich, Cat#C2181) and biotin-conjugated anti-rabbit IgG (1:500, Dianova, Cat#111-065-003) diluted in 2% NGS in PBS for 2 h. Subsequently, a solution containing DAPI (1:1000, Merck, Cat#32670) and Alexa Fluor 647-conjugated Streptavidin (1:1000, Thermofisher, Cat#S21374) diluted in 2% NGS in PBS was applied to the samples for 1 h. Finally, OBSCs were mounted on slides for confocal analysis.

#### 2.3.6. Confocal Imaging and Analysis

OBSCs were imaged using the Zeiss LSM 880 Airyscan confocal microscope (Carl Zeiss Microscopy GmbH, Jena, Germany). Images of the PI-labeled cells were acquired with a 10× air magnification objective, using DAPI as a reference. For astrocyte phagocytic activity in OBSCs, images were taken with a 63× oil objective. Six images per OBSC were acquired from the infralimbic/prelimbic areas of the PFC. Thirty z-stack images were taken at 0.50 µm intervals, using GFAP as a reference.

The intensity of PI+ cells weas quantified in the PFC using corrected total cell fluorescence (CTFC) on ImageJ, after subtracting the background noise. OBSC viability was analyzed after 0, 7, 14 and 21 days in culture (DIV). The mean CTFC value per brain was calculated and expressed as a percentage of the difference to DIV0. Analysis of astrocyte phagocytosis (“phagocytic index”) in OBSCs was carried out utilizing a different procedure than in pup brain due to the intrinsic nature of OBSCs which prevented us from selecting and examining single astrocytes. In brief, the analysis was performed on whole image areas for every picture with IMARIS 9.8 software (Bitplane, Zurich, Switzerland). A co-localization channel of LAMP1+ and synaptophysin+ was created and masked with the GFAP+ channel. Finally, astrocyte phagocytosis was determined as the ratio of “synphys+/LAMP1+/GFAP+” to “synphys+/LAMP1+” colocalized voxels.

### 2.4. Statistical Analysis

For statistical analysis, the normality of the distribution of data was verified before running the appropriate statistic tool. Data were analyzed via a one-way ANOVA for multiple comparisons. We used the ANOVA with repeated measures for OBSCs (with mixed-effects whenever missing values occurred), as slices derived from the same brain were analyzed at different time points. The analysis was followed by Tukey’s post hoc test and data are presented as the mean ± standard deviation (SD). Statistical significance was considered when the *p* value was equal or less than 5% (*p* < 0.05). All statistical analyses and data visualization were carried out using GraphPad Prism 8 (GraphPad Software, San Diego, CA, USA).

## 3. Results

### 3.1. MEGF10, PSD95 and Synaptophysin Expression in the PFC of Male and Female Developing Rat Pups

We first examined the differences in the expression of the phagocytic protein MEGF10 and the synaptic markers synaptophysin (synphys) and PSD95 in the cortex of male and female rat pups from P7 to P32. We observed that in males, MEGF10 total protein levels remained unaltered at P7–P21 and decreased at P32 (Figure 1B,C, one-way ANOVA, F(3,27) = 3.87, *p* < 0.0218, Tukey’s post hoc test: P21–P32, * *p* < 0.05). Contrarily, in female pups, MEGF10 total protein levels did not change at any of the developmental stages examined (Figure 1H,I, one-way ANOVA, F(3,24) = 1.399, *p* = 0.2673, ns, not significant).

We found additional differences between sexes when we evaluated the expression of synphys and PSD95. In males, we observed a significant rise in the amount of PSD95 between P7 and P14, reaching a peak and remaining elevated throughout the subsequent stages (Figure 1D,E, one-way ANOVA, F(3,27) = 9.714, *p* = 0.0002; Tukey’s post hoc test: P7–P14, * *p* < 0.05, P7–P21, *** *p* < 0.001, P7–P32, ** *p* < 0.01). In females, the levels of PSD95 were already slightly higher than in males at P7 and showed a peak of expression at P21, subsequently decreasing again by P32 (Figure 1J,K, one-way ANOVA, F(3,25) = 4.646, *p* = 0.0103; Tukey’s post hoc test: P7–P21, * *p* < 0.05, P14–P21, * *p* < 0.05).

When we examined synphys, we observed that, in males, changes in the expression levels mirrored a similar pattern to PSD95, with a rise between P7 and P14, which plateaued throughout P21 and P32 (Figure 1F,G, one-way ANOVA, F(3,26) = 13.28, *p* < 0.0001; Tukey’s post hoc test: P7–P14, * *p* < 0.05, P7–P21, **** *p* < 0.0001, P7-P32, *** *p* < 0.001). In females, the increase in protein expression followed a delayed pattern similar to PSD95, with a peak at P21, which was still slightly elevated at P32, although less prominently than at P21 (Figure 1L,M, one-way ANOVA, F(3,24) = 11.82, *p* < 0.0001; Tukey’s post hoc test: P7–P21, **** *p* < 0.0001, P14–P21, ** *p* < 0.001, P7–P32, * *p* < 0.05).

### 3.2. Astrocyte-Dependent Synapse Elimination in the PFC of Male and Female Developing Rat Pups

Although the markers for adult astrocytes have been identified and successfully used so far [16], it is still under debate which markers can unequivocally label early-stage postnatal astrocytes. Based on the work of Raponi and colleagues [17], we decided to use a combination of GFAP and S100ß antibodies to identify early-stage mature astrocytes of the PFC. We performed immunofluorescent immunohistochemistry in brain slices and quantified the phagocytic index (for details, see Section 2.2.3 of the Materials and Methods Section) using antibodies against synaptophysin and lysosomal-associated membrane protein 1 (LAMP1) (Figure 2B–G). This analysis was specifically carried out in the infralimbic/prelimbic areas of the PFC in male and female pups from P7 to P32 (Figure 2A).

When assessing the changes in the rates of synapse elimination, we observed sex-dependent differences again. In males, we measured a rise in synapse elimination between P7 and P14, followed by a decrease at P21 and a further drop at P32 (Figure 2C,D, one-way ANOVA, F(3,12) = 17.17, *p* = 0.0001; Tukey’s post hoc test: P7–P14, * *p* < 0.05, P7–P32, * *p* < 0.05, P14–P21, ** *p* < 0.001, P14–P32, **** *p* < 0.0001). In females, the pattern of synapse elimination displayed higher levels of the phagocytic index, already detectable at P7, which remained elevated at P14, went down at P21 and slightly increased again at P32 (Figure 2F,G, one-way ANOVA, F(3,12) = 11.12, *p* = 0.0009; Tukey’s post hoc test: P7–P21, ** *p* < 0.001, P14–P21, ** *p* < 0.001).

To characterize the changes in the synphys protein with greater precision and localization than previously achieved via Western Blot, we analyzed the synphys fluorescence intensity in the infralimbic/prelimbic areas of the PFC (Appendix A). This analysis revealed that, in males, a rapid increase in the signal occurred between P7 and P14, followed by an equally rapid signal decline by P21, which stabilized at P32, mirroring the respective patterns of synapse elimination (Appendix A, one-way ANOVA, F(3,12) = 21.36, *p* < 0.0001; Tukey’s post hoc test: P7-P14, **** *p* < 0.0001, P14–P21, ** *p* < 0.001, P14–P32, *** *p* < 0.001). Similar to males, the synphys fluorescent intensity in females exhibited a rapid rise from P7 to P14 and a decline between P14 and P21. However, in contrast to males, the signal increased again by P32, suggesting that a second wave of synaptogenesis may occur in female pups (Appendix A, one-way ANOVA, F(3,12) = 18.01, *p* < 0.0001; Tukey’s post hoc test: P7–P14, ** *p* < 0.01, P14–P21, **** *p* < 0.0001, P14–P32, * *p* < 0.05, P21–P32, * *p* < 0.05). Remarkably, however, in females, these changes in synaptogenesis did not mirror the respective patterns of synapse elimination.

### 3.3. Establishing Organotypic Brain Slice Cultures (OBSCs) to Examine Synapse Elimination in the PFC during Early Postnatal Developmental Stages

#### 3.3.1. Viability Assay

The procedure used to prepare organotypic slices may cause tissue damage, which in turn has the potential to affect their viability (33). To assess the amount of cellular damage over the culturing period, OBSCs were labeled with propidium iodide (PI), which is only able to penetrate compromised cell membranes and therefore marks dead or dying cells. DAPI was additionally used to counterstain cell nuclei and precisely identify single cells. The intensity of PI staining was then used as a correlate measure of cell death, and data from 7-day in vitro (DIV7) to DIV21 samples were normalized to DIV0, the time point taken immediately after cutting.

We could observe an increase in the PI staining intensity from DIV0 to DIV7 (Figure 3), which remained stable afterward (Figure 3A; one-way ANOVA, F(3,13) = 5.984; *p* = 0.0086), thereby suggesting that the viability of the slices was maintained along all experimental stages.

#### 3.3.2. MEGF10 Expression and Astrocyte-Dependent Synapse Elimination in the PFC of OBSCs Derived from Male and Female Rat Pups

To validate OBSCs for investigating the differences in astrocyte-mediated age- and sex-dependent phagocytosis, we first analyzed MEGF10 expression levels in the PFC of OBSCs prepared from male and female rat pups at DIV7, 14 and 21.

In accordance with the results obtained from rat pups (Figure 1B,C,H,I), we observed no differences in the expression of MEGF10 levels in male OBSCs between DIV7 and DIV21 (Figure 4A,B, one-way ANOVA with mixed-effects model, F(2,11) = 2.563, ns, not significant). The same trend was observed for female OBSCs, which also showed no detectable differences at any of the stages examined (Figure 4E,F, one-way ANOVA with repeated measures, F(2,9) = 1.937, ns, not significant).

Subsequently, we evaluated astrocyte-dependent phagocytosis using GFAP to label astrocytes. This approach was chosen to avoid difficulties in the analysis of single astrocytes in this type of ex vivo system, where double staining with S100ß might hinder the identification and examination of single cells (Ref. [44] and our data). This experiment revealed that in OBSCs derived from male rat pups, the rate of astrocyte-mediated synapse elimination (measured as described in Section 2.3.5 of the Materials and Methods section) increased between DIV7 and DIV14 to reach a peak and drop down again at DIV21, as we previously observed in rat pups at comparable developmental stages (Figure 4C,D, one-way ANOVA with repeated measures, F(2,6) = 11.14, *p* = 0.0096, Tukey’s post hoc test: DIV7-DIV14, * *p* < 0.05, DIV14–DIV21, ** *p* < 0.001). In female OBSCs, however, differently than from female rat pups, we observed an analogous trend in changes as in males, with a significant increase in the rate of phagocytosis between DIV7 and DIV14, which declined again between DIV14 and DIV21 (Figure 4G,H, one-way ANOVA with repeated measures, F(2,6) = 8.067, *p* = 0.0199, Tukey’s post hoc test: DIV7–DIV14, * *p* < 0.05, DIV14–DIV21, *p* = 0.0795).

We further examined temporal changes in synphys expression in OBSCs. In contrast to the results from pup brains, in OBSCs, the signal fluorescence intensity displayed a higher degree of variability, with no significant changes at any time point in neither males nor females (Appendix A, males, one-way ANOVA, F(2,6) = 0.52, ns, not significant; Appendix A, females, one-way ANOVA, F(2,6) = 0.09, ns, not significant).

## 4. Discussion

Postmortem studies in human and non-human primates have shown that in cortical areas all synaptogenic events leading to supernumerary synapses and the subsequent age-dependent synapse elimination, increase after birth. These processes reach a peak in early childhood and decline during later developmental stages, finally becoming refined in late adolescence/early adulthood [33,45,46,47,48,49]. However, brain scan imaging methods have shown that dynamic changes in the density of the gray matter may persist longer than adolescence, opening the question of whether such events continue beyond it and extend into the third decade of life, before reaching adult levels [50]. Such long developmental range of neuronal network reorganization could also account for the substantial impact of environmental challenges on the formation of human cognitive and emotional capacities, as well as their potent detrimental effects, which may underlie the onset of neuropsychiatric disorders [32].

In addition to these intrinsic neurobiological modifications, it is also clear that sex-specific hormonal changes may influence synaptic formation/elimination to favor the development of adaptive behavioral responses during sexual maturation in both males and females. However, only few studies consider sex differences among the critical factors that should be evaluated when interpreting experimental findings and measured parameters [34,48]. Therefore, research in this direction is highly warranted to better understand the sex-dependent maturational trajectories of brain development within healthy contexts, which can in turn guide medical interventions in disease states with a focus on sex differences.

Here, we sought to offer an alternative tool to examine age- and sex-dependent changes in astrocyte-mediated synapse elimination in the prefrontal cortex (PFC) at early postnatal stages, from birth until adolescence. We compared developing brains from rat pups with organotypic brain slices (OBSCs), a 3D system so far used to investigate hippocampal and cerebellar developmental processes [37,42,44,51,52,53,54]. In addition, this system is more amenable to pharmacological, genetic and various manipulations than other models and may substantially reduce eperimental costs. In rat pups, we examined the age- and sex-dependent rates of synapse formation and elimination, and evaluated whether putative sex-dependent differences in astrocyte-mediated synapse elimination occurred. We found that MEGF10 showed slight oscillations in the average of its protein levels between P21 and P32 in males (Figure 1B,C), which marks the end of the juvenile period and corresponds to the generally accepted time for the closure of the cortical developmental critical period [55]. On the contrary, MEGF10 levels did not change in females at any developmental stage (Figure 1H,I). These findings were effectively reproducible in OBSCs, which did not show any relevant modifications in the levels of MEGF10 between DIV7 and DIV21 (Figure 4A,B,E,F). While these results might initially suggest a lack of MEGF10-dependent phagocytic activity from P7 to P32, our observations were in apparent contradiction to such expectations. We measured both age- and sex-dependent changes in astrocyte-mediated synaptic phagocytosis (Figure 2B–G), indicating the different mechanisms regulating MEGF10 activity compared to changes in its expression levels. For instance, it has been shown that for the MEGF10-mediated uptake of amyloid-ß plaques, the role of a lipid raft-dependent endocytosis, in the absence of any changes in MEGF10 protein levels, is relevant for MEGF10 functionality [56]. Further work is needed to clarify whether this possibility applies to OBSCs.

To next evaluate whether this system represented a useful platform to examine changes in synaptic elimination during the critical period of cortical development, we first measured and correlated changes in synaptic proteins with the astrocyte-mediated phagocytosis in pups from P7 to P32.

Here, we observed clear age- but also sex-dependent developmental patterns, which suggested that sex-specific determinants influence both the rates of synaptogenesis and astrocyte-mediated synapse elimination (Figure 1D–G,J–M and Figure 2B–G).

We initially detected an apparent inconsistency between the results obtained for the expression levels of synaptophysin via Western blot (Figure 1F–G,L,M) and immunohistochemistry (Appendix A). On the one hand, the expression of this protein in Western blot increased from P7 to P14, remaining elevated until P32 in males (Figure 1F,G), while it peaked with a delayed increase in females between P14 and P21 (Figure 1L,M). On the other hand, the IHC showed a different pattern, with an apparent peak at P14 which decreased afterward in males, whereas it decreased in females at P21, only to increase again by P32 (Figure 2B,E and Appendix A). The observed discrepancies could be attributed to distinct methods used to prepare tissues for the two experiments. As mentioned in the Section 2 of Methods section, the brain tissue for Western blot analysis might have contained tissue from other brain cortical areas around the PFC (Figure 1A), whereas the IHC analysis was restricted to the infralimbic/prelimbic areas of the PFC (Figure 2A). Achieving complete congruence within experimental conditions, such as using laser microdissection, might be necessary to address these differences precisely.

Our results for the pups showed that at the end of the juvenile period, at around P32, the pattern of astrocyte-dependent phagocytosis also followed a different trend in females when compared to males in the PFC (Figure 2B–G). We observed a second slight rise in the rate of synapse pruning at P32, which correlated with a significant elevation in synaptophysin levels only in females at this time point (Appendix A). These results suggest that in females, multiple waves of synaptogenesis and synaptic refinement via astrocyte-dependent synapse elimination might occur (at P7/P14 and at P32). This phenomenon could also be examined in human and non-human primates to possibly explain the observed reduced sizes of the PFC in adult females compared to males [34,48]. However, further experiments should be conducted to validate this hypothesis, including the examination of later developmental stages and, if possible, increasing samples sizes.

From our results, we could argue that a detrimental environmental hit, such as stress or a traumatic event, during late adolescence in females may have long-lasting effects, as the system appears to be still in the developmental phase with respect to the male counterparts. This is in line with observations in disorders more prevalent in females, which may be linked to the different maturational trajectories of the neuronal circuits between the two sexes.

These differences could depend on long-range synaptic innervation, but also local cellular mechanisms, which might both contribute to the sex-specific synaptic refinements in the infralimbic/prelimbic areas of the PFC. Therefore, we established a 3D culturing system to examine developmentally regulated, but also sex-dependent mechanisms, which control synaptic elimination during the juvenile stages of postnatal brain growth. As discussed before, the comparison of the early PND stages of pups’ brain growth with OBSCs revealed that slices prepared from P4 to P6, a period when sensory inputs have already reached cortical areas [57] and astrocyte proliferation/differentiation processes are primarily local [16], might retain sufficient autonomous information to reproduce an in vivo brain environment. This allows for astrocytes to display very similar patterns of MEGF10 expression in the cortex of both sexes in pups and OBSCs (Figure 1B,C,H,I and Figure 4A,B,E,F). More strikingly, we noticed that in males, astrocyte-mediated synaptic phagocytosis was preserved in OBSCs and followed the same dynamics showed by the developing brains (Figure 2B–D and Figure 4C,D). These first results suggested that OBSCs might undergo a sort of developmentally programmed ‘reset’ after cutting, and show similarities between the P7 and DIV7 stages. However, the findings were notably different in female OBSCs, with the change in the rate of phagocytosis resembling male-derived OBSCs, with a peak at DIV14 and reduction afterward. This was in contrast to the female developing brains, where the levels of phagocytosis at P7 were as high as those at P14 and dropped down at P21 (Figure 2E–G and Figure 4G,H). These results suggest that this parameter is not sufficiently preserved yet in our model system to confidently claim its validity for the examination of female-specific changes in astrocyte-dependent synapse elimination. We hypothesized that this factor may be affected by the cutting procedure [40], which might hinder essential long-range, most likely peripheral, sex-specific determinants, such as sexual hormones, from reaching the cortex and influencing the sex-dependent reshaping of this distinct astrocyte-mediated function. More experiments should be performed to assess whether interventions, such as the administration of sexual hormones in OBSCs, might help to phenocopy the phagocytic rate found in the developing female brains. However, even though cultures can be maintained for long periods [40,54], studies of hormonal interferences with neuronal network formation in this system may have limitations.

Another possible explanation for the observed differences could be related to the method we used to quantify the rate of synaptic pruning in OBSCs, since the immunohistochemical staining employed for the brains had to be slightly adapted for the slices to allow for the characterization of astrocyte-associated synaptic engulfment. To rule out the possible confounding factors, the use of virally mediated astrocyte labeling in OBSCs may help in refining the characterization of the rates of synaptic material engulfment in this system. However, this procedure may reintroduce a high inter-sample variability and has to be carefully established to reach an acceptable degree of reproducibility and reliability of experimental findings.

Finally, in view of the acknowledged limitations of OBSCs for conducting studies on synaptic changes in cortical areas because of the disruption of both short- and long-range axonal inputs during the cutting procedure [40], we expected reduced rates of synaptic elimination in the brain slices. Moreover, we were aware of the possible increase in astrocyte gliosis, which has been described in OBSCs and might have affected our results [58]. However, it has been reported that lysosomal-dependent phagocytosis would especially be reduced and not increased in reactive astrocytes [59]. This further reinforces the specificity of our results obtained in OBSCs on the dynamically changing rates of phagocytosis in the PFC along PND stages and lack of artifactual effects resulting from tissue damage during dissection.

Of course, other limitations exist, such as the impaired capability of directly correlating any type of induced manipulation with complex changes, such as behavioral parameters.

However, overall, this system offers numerous advantages which should be considered when planning animal research experiments. For example, it allows for the direct correlation of the effects with the selected molecular or biochemical processes or cell-type-specific responses occurring in situ in distinct brain regions. This is surely also in line with animal welfare regulations, which encourage the use of alternative model systems to sensibly reduce the number(s) of animals used to perform multiple experiments and measure various parameters in biomedical research, without compromising the acquisition of strong and reliable data from molecular or histochemical examinations [60].

Here, we demonstrated that OBSCs can serve as a useful supporting tool to examine age- and sex-dependent changes in astrocyte-dependent synapse elimination in the rat PFC during postnatal developmental stages. This model might be very valuable for particularly evaluating male-related differences in astrocyte responses to environmental or endogenous insults, which may induce synaptic aberrancies. Therefore, OBSCs may prove to be an ideal model to study the neurobiological underpinnings of neuropsychiatric or neurodevelopmental disorders characterized by synaptic deficits correlated with astrocyte pathology, such as schizophrenia, autism disorder or major depressive disorder.

## Figures and Tables

**Figure 1 cells-12-02761-f001:**
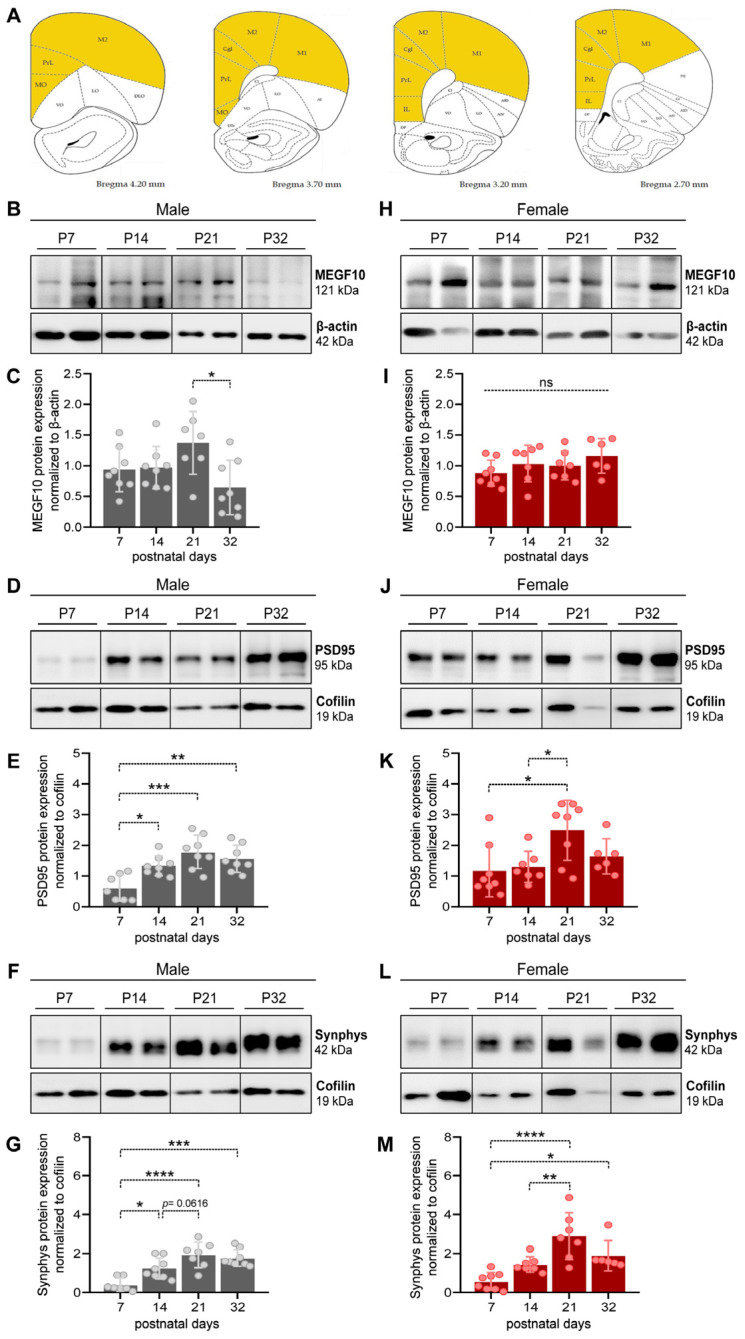
Progressive expression of MEGF10 and synaptic markers during the cortical critical period in male and female littermates. (**A**) Representative graphs from the Rat Brain Atlas to show the areas of tissue dissected for Western blots. (**B**,**D**,**F**) For males, representative lanes were cropped from immunoblots to show changes in MEGF10 (**B**) PSD95 (**D**) and synaptophysin (**F**) from isolated cortices of male pups at postnatal days (P) 7, 14, 21 and 32. Full blots are shown in Appendix A. (**C**,**E**,**G**) Quantitative analysis of total MEGF10 (**C**), PSD95 (**E**) and synaptophysin (**G**) protein expression normalized to β-actin (for MEGF10) and cofilin (for PSD95 and synaptophysin) in male pups at different developmental stages. Each dot represents one animal. Statistical analysis was performed via a one-way ANOVA and Tukey’s multiple comparison. ns, not significant; * *p* < 0.05; ** *p* < 0.01; *** *p* < 0.001; **** *p* < 0.0001. (**H**,**J**,**L**) For females, representative lanes were cropped from immunoblots of MEGF10 (**H**), PSD95 (**J**) and synaptophysin (**L**) from isolated PFC of female pups at postnatal days (P) 7, 14, 21 and 32. Full blots are shown in Appendix A. (**I**,**K**,**M**) Quantitative analysis of total MEGF10 (**I**), PSD95 (**K**) and synaptophysin (**M**) protein expression normalized to β-actin (for MEGF10) and cofilin (for PSD95 and synaptophysin) in female pups at different developmental stages. Each dot represents one animal. Statistical analysis was performed via one-way ANOVA and Tukey’s multiple comparison. ns, not significant; * *p* < 0.05; ** *p* < 0.01; *** *p* < 0.001; **** *p* < 0.0001. All data are presented as the mean ± SD.

**Figure 2 cells-12-02761-f002:**
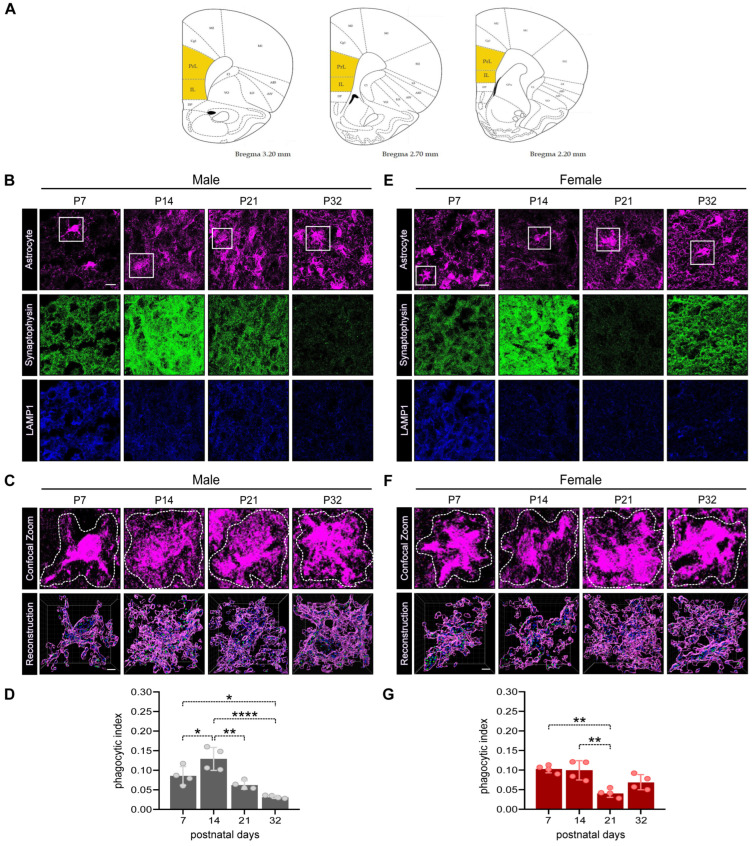
Astrocyte-mediated synaptic pruning during developmental stages in male and female infralimbic/prelimbic areas of the PFC. (**A**) Representative graphs from the Rat Brain Atlas (46) to show areas of tissue labelled for immunofluorescent immunohistochemistry and confocal imaging. (**B**) Representative confocal images of brain slices from male littermates labelled with GFAP/S100ß (astrocytes, magenta), synaptophysin (green) and LAMP1 (blue) at P7, 14, 21 and 32. Scale bar 10 µm. (**C**) Representative confocal (upper panels) and Imaris surface-rendered (lower panels) images of analyzed male astrocytes (insets from (**B**)). In the 3D reconstructions, only LAMP1+ and synaptophysin+ spots inside the astrocyte volume are rendered. Scale bar 2 µm. (**D**) Quantification of the engulfed synaptophysin spots within LAMP1 spots in astrocytes of the PFC normalized to the astrocyte volume at different postnatal developmental stages in male littermates. Each dot represents the average data of 6 analyzed astrocytes from each animal: n = 4 animals. One-way ANOVA, Tukey’s multiple comparison, * *p* < 0.05; ** *p* < 0.01; **** *p* < 0.0001. (**E**) Representative confocal images of brain slices from female littermates labelled with GFAP/S100ß (astrocytes, magenta), synaptophysin (green) and LAMP1 (blue) at P7, 14, 21 and 32. Scale bar 10 µm. (**F**) Representative confocal (upper panels) and Imaris surface-rendered (lower panels) images of analyzed female astrocytes (insets from (**E**)). In the 3D reconstructions, only LAMP1+ and synaptophysin+ spots inside the astrocyte volume are rendered. Scale bar 2 µm. (**G**) Quantification of the engulfed synaptophysin spots within LAMP1 spots in astrocytes of the PFC normalized to the astrocyte volume at different postnatal developmental stages of female littermates. Each dot represents the average data of six analyzed astrocytes from each animal: n = 4 animals. One-way ANOVA, Tukey’s multiple comparison, ** *p* < 0.01. All data are presented as the mean ± SD.

**Figure 3 cells-12-02761-f003:**
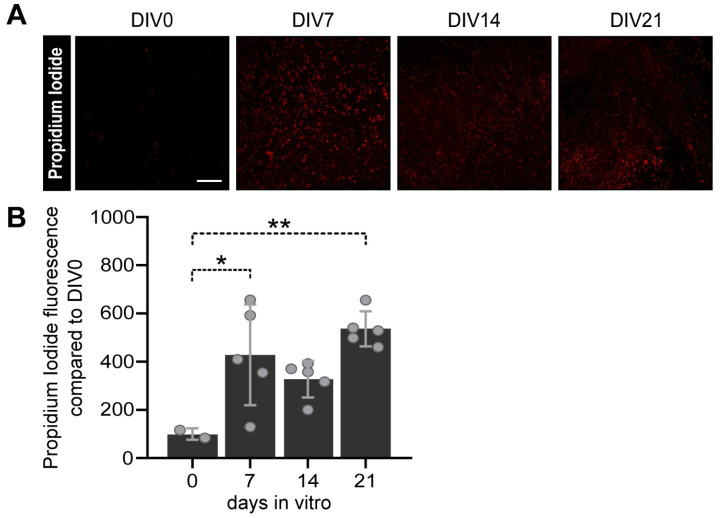
Assessment of cell viability in organotypic brain slice culture (OBSCs). (**A**) Representative confocal images of selective cell death (red) in the prelimbic/limbic area of OBSCs at different timepoints: days in vitro (DIV) 0, 7, 14 and 21. Scale bar 200 µm. (**B**) Quantification of cell death assessed via propidium iodide (PI) total fluorescence intensity normalized to DIV0. Each dot represents a single animal. Statistical analysis was performed via a one-way ANOVA and Tukey’s multiple comparison, * *p* < 0.05; ** *p* < 0.01. Data are presented as the mean ± SD.

**Figure 4 cells-12-02761-f004:**
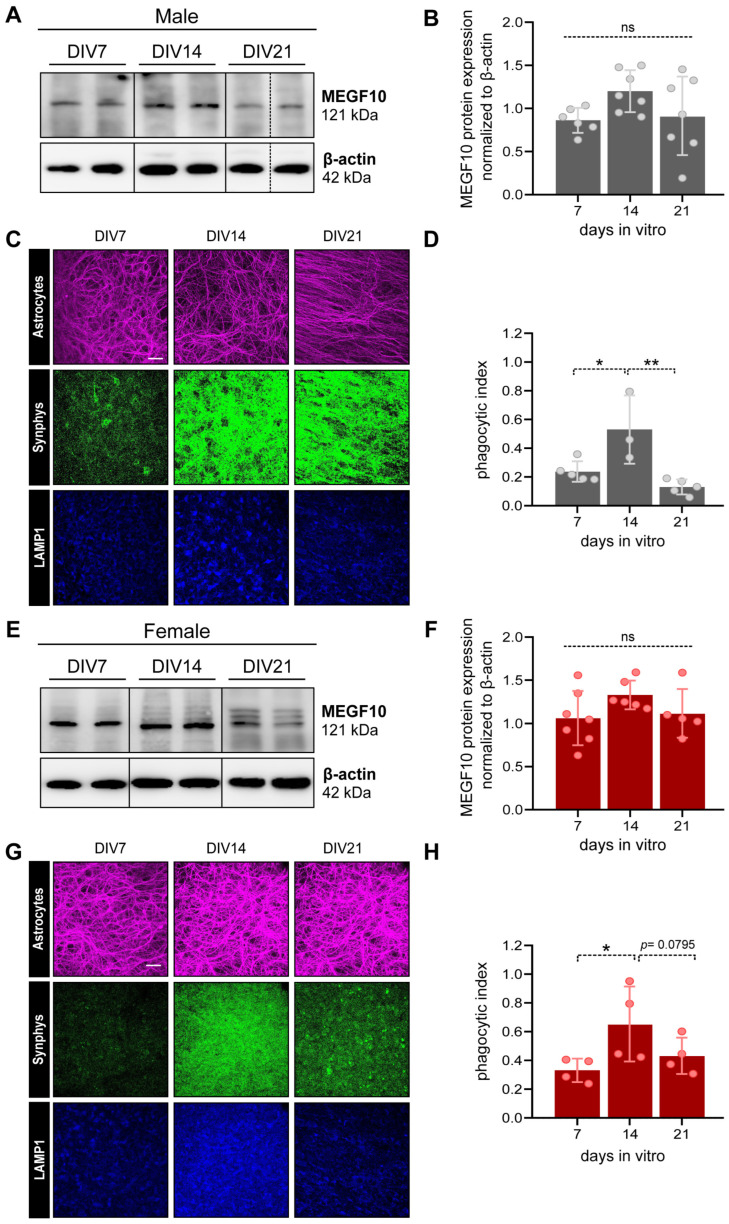
MEGF10 expression in the cortex and astrocyte phagocytic capacity in infralimbic/prelimbic areas of the PFC of male and female OBSCs. (**A**) Representative lanes cropped from immunoblots show MEGF10 protein expression in the cortex isolated from male-derived OBSCs at DIV7, 14 and 21. Full blots are shown in Appendix A. (**B**) Quantitative analysis of total MEGF10 protein expression normalized to β-actin in male-derived OBSCs at different timepoints. Each dot represents one animal, n = 6–7 animals. Statistical analysis was performed via one-way ANOVA repeated measures, with Tukey’s multiple comparison, ns, not significant. (**C**) Representative confocal images of OBSCs labelled with GFAP (astrocytes, magenta), synaptophysin (green) and LAMP1 (blue) at DIV7, 14 and 21 in male-derived OBSCs. Scale bar 25 µm. (**D**) Quantification of co-localized voxels positive for LAMP1, synaptophysin and GFAP. The phagocytic index is expressed as the ratio of “synphys+/LAMP1+/GFAP+” to “synphys+/LAMP1+” colocalized voxels. Each dot represents the average data of five pictures from one animal, n = 3–5 animals. Statistical analysis was performed via one-way ANOVA repeated measures (mixed effects) and Tukey’s multiple comparison, * *p* < 0.05; ** *p* < 0.01. All data are presented as the mean ± SD. (**E**) Representative lanes cropped from immunoblots show MEGF10 protein expression in the cortex isolated from female-derived OBSCs at DIV7, 14 and 21. Full blots are shown in Appendix A. (**F**) Quantitative analysis of total MEGF10 protein expression normalized to β-actin in female-derived OBSCs at different timepoints. Each dot represents one animal, n = 5–7 animals. Statistical analysis was performed via one-way ANOVA repeated measures and Tukey’s multiple comparison, ns, not significant. (**G**) Representative confocal images of OBSCs labelled with GFAP (astrocytes, magenta), synaptophysin (green) and LAMP1 (blue) at DIV7, 14 and 21 in female-derived OBSCs. Scale bar 25 µm. (**H**) Quantification of co-localized voxels positive for LAMP1, synaptophysin and GFAP. The phagocytic index is expressed as the ratio of “synphys+/LAMP1+/GFAP+” to “synphys+/LAMP1+” colocalized voxels. Each dot represents the average data of five pictures from each animal, n = 4 animals. Statistical analysis was performed via one-way ANOVA repeated measures and Tukey’s multiple comparison, * *p* < 0.05; trend, *p* = 0.0795. All data are presented as the mean ± SD.

## Data Availability

The data are contained in the article and Appendix A.

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
