# Peer review of "Early Age- and Sex-Dependent Regulation of Astrocyte-Mediated Glutamatergic Synapse Elimination in the Rat Prefrontal Cortex: Establishing an Organotypic Brain Slice Culture Investigating Tool"

_cells, 2023, doi:10.3390/cells12232761_

Round 1

Reviewer 1 Report

Comments and Suggestions for Authors

Vivi E. et al. tried to establish an Organotypic Brain Slice Culture 18 (OBSC) system to more efficiently investigating early brain maturational trajectories. The idea is cool but a lot of your data do not convince to me.

1.     As you described that astrocyte-dependent synapse elimination (pruning) mediated by the MEGF10 phagocytic pathway. Both your in vivo and ex vivo show that MEGF10 has no significant changes before P21 and DIV21 and the phagocytic index has significant changes.

2.     Your synphys express showed different pattern between western-blot and IHC during development.

3.     It is interesting that you showed synphys positive synaptogenesis pattern very similar to synapse pruning pattern (phagocytic index). Do you mean that pruning happened with synaptogenesis together? It doesn’t make sense to me.

4.     In your figure 2G, you interestingly showed that synaphy Fl intensity was significantly increased in P32. How to explain it.

5.     Your western-blot data have such big vibration. I guess that you did not loading the same amount of protein samples on gel.

6.  I will suggest much more detail describe how to measure the phagocytic index in methods and figure legend. And provide more reference for this method, if had.

Author Response

  1. As you described that astrocyte-dependent synapse elimination (pruning) mediated by the MEGF10 phagocytic pathway. Both your in vivo and ex vivo show that MEGF10 has no significant changes before P21 and DIV21 and the phagocytic index has significant changes.
  • Thank you for your comment, indeed we acknowledge an apparent contradiction in our results. However, it is important to note that protein activity is regulated not only by their concentrations but also their localization. In this specific case, MEGF10 localization in/out the lipid rafts has been established as a determinant of its endocytotic activity (see Singh et al, 2010). Therefore a lack of difference in its concentration might not necessarily reflect its functionality or lack thereof. In other experiments, that we may confidentially provide upon request (part of another manuscript in preparation), we in fact observe that the lack of astrocyte-mediated MEGF10-dependent phagocytosis is not attributed to a reduced MEGF10 expression, but rather to its mislocalization outside lipid rafts. Moreover, lipid rafts disruption in intact astrocytes is sufficient to fully mimic an impaired astrocyte-mediated phagocytosis to the extent as MEGF10 downregulation does.
  • To consider your comment and clarify putative concerns of the readers, we have included a sentence in the Discussion:”While these results might initially suggest a lack of MEGF10-dependent phagocytic activ-ity from P7 to P32, our observations were in apparent contradiction to such expectations. We measured both age- and sex-dependent changes in astrocyte-mediated synaptic phagocytosis (Fig 2B-G), indicating different mechanisms regulating MEGF10 activity compared to changes in its expression levels. For instance, it has been shown that for the MEGF10-mediated uptake of amyloid-ß plaques, the role of a lipid raft-dependent endo-cytosis, in the absence of any changes in MEGF10 protein levels, is relevant for MEGF10 functionality [46]. Further work would be needed to clarify whether this possibility applies to OBSCs”. (lines 536-544).
  1. Your synphys express showed different pattern between western-blot and IHC during development.
  • Thank you for bringing this to our attention. Indeed, the variability observed aligns with a recognized issue that may occur with western blot procedures, as WB measures total levels of denatured proteins from various cells/cellular compartments, whereas IHC detects native proteins localized in selected cellular compartments in situ. In line with this, it is crucial to note that changes in protein conformation occurring during denaturation can lead to potentially misleading outcomes. We also mentioned that we observed such differences, highly likely arising from the alternative preparation methods: for WB, we dissect brain tissues, trying to take only the prefrontal areas, albeit with the potential inclusion of adjacent tissue (lines 126-129:”Whenever possible, mostly prefrontal cortex (PFC) tissue was collected at different postnatal stages (P0, P7, P14, P21, P32)….. However, tissue extracted might include proteins from neighboring areas (Fig 1A)”. With IHC we can examine more localized brain areas and therefore identify subtle differences not detectable with WB (lines 372-374:” To characterize changes in synphys protein with greater precision and localization than previously achieved with Western Blot, we analyzed synphys fluorescence intensity in the infralimbic/prelimbic areas of the PFC (Suppl Fig 2A, B).”
  1. It is interesting that you showed synphys positive synaptogenesis pattern very similar to synapse pruning pattern (phagocytic index). Do you mean that pruning happened with synaptogenesis together? It doesn’t make sense to me.
  • Thank you for this observation. In fact, this appears to be only a correlative event happening in male brain pups, but not neither in female pups (lines 384-385 “Remarkably, however, in females these changes in synaptogenesis did not mirror the respective patterns of synapse elimination”) nor in OBSCs, where a notably higher degree of variability in synaptogenesis was observed. Despite this variability, specific age-related oscillations in phagocytic activity were evident (reported in lines 450-461). This suggests that the relationship between the differences in rates of synaptogenesis and synaptic pruning might not be necessarily causative. However, it is important to note that in OBSC the absence of long-range synaptic innervations is a critical factor. This absence may induce an increase in phagocytic activity to regulate the amount of synapses which should be maintained versus those which should be eliminated. However, additional research efforts would be required  to clarify this aspect, but this  extends beyond the current scope of our study.
  1. In your figure 2G, you interestingly showed that synaphy Fl intensity was significantly increased in P32. How to explain it.
  • Indeed, we also found this result interesting and commented on it in our discussion (lines 566-576, yellow-marked).
  1. Your western-blot data have such big vibration. I guess that you did not loading the same amount of protein samples on gel.
  • We actually aimed to load consistent amounts of total proteins, as stated in the Matherial and Methods (line 136-137: “A total amount of 12µg of protein per lane was separated by electrophoresis and transferred to nitrocellulose membranes”). However, technical variations might have occurred. In order to mitigate the risk of misinterpretations, we additionally used internal loading controls (cofilin or ß-actin) to normalize measured protein. This normalization strategy allowed for the comparison of relative differences among PND stages with greater accuracy.
  • To incorporate your helpful comment, we have now added a sentence in the Materials and Methods to increase awareness of pitfalls of our procedures for the readers in lines 138-142.
  1. I will suggest much more detail describe how to measure the phagocytic index in methods and figure legend. And provide more reference for this method, if had.
  • We have added more details about the procedure (lines 197-205) and we hope that it is clearer now. The procedure has not been published before in astrocytes, but has been adapted from protocols used to measure phagocytic activity in microglia cells.

Reviewer 2 Report

Comments and Suggestions for Authors

In the present study, the authors investigated age- and sex-dependent astrocyte-mediated synapse elimination (pruning) in organotypic brain slice cultures (OBSCs) of rat prefrontal cortex and compared the results with those in vivo. They concluded that OBSCs can be a valid model to study physiological and pathological astrocyte-mediated synaptic remodeling although further optimizations are needed to use OBSCs to reveal sex-dependent differences. Although this study demonstrated a possibility that OBSCs can be used for a study of astrocyte-mediated synaptic remodeling, as the authors concluded, the following portions in the present manuscript need to be addressed for publication in Cells. 

Major:

1. I cannot understand "phagocytic index" clearly. 

1-1. "Phagocytic index" in vivo is "normalized to the astrocyte volume". How is "the astrocyte volume" calculated? 

1-2. Is calculation of "phagocytic index" same or different between in vivo and OBSCs? 

1-3. The number/density of astrocytes may increase developmentally. In addition, the number/density of astrocytes may increase owing to gliosis in OBSCs. Is "phagocytic index" affected by these changes in the number/density of astrocytes? 

2. The authors showed the temporal changes in the amount of astrocyte-mediated synapse elimination in vivo and in OBSCs. However, how are the temporal changes in the amount of the whole (not only "astrocyte-mediated" but also other types of) synapse elimination (synaptophysin+/LAMP1+ spots) in vivo and in OBSCs? 

3. Although the temporal changes in synaptophysin expression in vivo are shown in Figure 2, those in OBSCs are not shown in Figure 4. How are the temporal changes in synaptophysin expression in OBSCs? 

4. page 14, lines 479-481: The hypothesis of "developmental programmed reset after cutting" by the authors is very interesting but was not proved at all. Is there any evidence for this hypothesis? For example, if brain slices are cut at P0 or P7, do both slices show the same temporal changes in MEGF10 expression and phagocytic index? 

Minor:

1. It is better to explain about MEGF10 in a little more detail in Introduction. 

2. Materials and Methods 2.2.2 and 2.3.5: The expression "tertiary antibody solution" is inadequate because this solution does not contain antibody. 

  In addition, this solution contains Avidin. However, I cannot find the result of staining with Avidin. 

3. page 13, lines 468-469: Although the authors wrote "so far used to investigate hippocampal developmental processes", the use of OBSCs is not limited to the hippocampus. In particular, references of investigations with cerebellar OBSCs are lacking. 

Comments on the Quality of English Language

English can be polished throughout the manuscript. 

Author Response

Major:

  1. I cannot understand "phagocytic index" clearly. 

1-1. "Phagocytic index" in vivo is "normalized to the astrocyte volume". How is "the astrocyte volume" calculated? 

  • The volume of single astrocytes was measured using the function ‘surface rendering’ of Imaris after 3D cell reconstruction. In the Method section, we have now changed the description of how we calculated the phagocytic index (adapted from Methods used to calculate it in microglia cells), lines 197-205.

1-2. Is calculation of "phagocytic index" same or different between in vivo and OBSCs? 

  • The phagocytic index could not be calculated in the exact same way between experiments, as elucidated in the sentence added in the Methods (lines 285-288):”Analysis of astrocyte phagocytosis (“phagocytic index”) in OBSCs was carried out with a different procedure than in brain pups due to the intrinsic nature of OBSC which prevented us from selecting and examining single astrocytes. In brief, the analysis was per-formed on whole image areas for every picture with IMARIS 9.8 software (Bitplane, Zurich, Swit-zerland). A co-localization channel of LAMP1+ and synaptophysin+ was created and masked with the GFAP+ channel. Finally, astrocyte phagocytosis was determined as ratio of “synphys+/LAMP1+/GFAP+” and “synphys+/LAMP1+” colocalized voxels.”
  • We have mentioned this aspect among the putative explanations on the differences in sex-dependent results observed between in vivo and OBSCs, which might in fact rely on the quantification method used in the two systems. We proposed an improvement to overcome this point in the disciussion lines 590-597:”….on the method we used to quantify the rate of synaptic pruning in OBSCs, because the immunohistochemical stainings used for brains had to be slightly adapted in slices to allow the characterization of astrocyte-associated synaptic engulfment. To rule out possible confounding factors, the use of virally-mediated astrocyte labelling in OBSCs may help to refine the characterizion of rates of synaptic material engulfment in this system. However, this procedure may increase again the inter-sample variability and has to be carefully established to reach high reproducibility and reliability of experimental findings.”

1-3. The number/density of astrocytes may increase developmentally. In addition, the number/density of astrocytes may increase owing to gliosis in OBSCs. Is "phagocytic index" affected by these changes in the number/density of astrocytes? 

  • Thank you for addressing this concern. Indeed, such age-dependent changes in number/density of astrocytes occur, our analysis method has been designed to account for these variations and mitigate potential biases. We have now revised the last sentence in Method lines 208-211 to answer your concern and hopefully clarify this issue for potential readers:”Moreover, the same number of astrocytes (six) per animal was analyzed at any given developmental stage to enable the comparison of age-dependent changes in phagocytic in-dex, independently of developmentally-driven increases in total numbers of astrocytes along PND stages”
  • With regard to the gliosis, we did not measure whether changes in number/density of total astrocytes might be an issue here. However, as mentioned before, our quantification method is designed to address this potential concern. We are aware that gliosis may affect the lysosomal pathway and we discussed it in lines 624-627: “Moreover, we were aware of the possible increase in astrocyte gliosis, which has been de-scribed in OBSCs and might have affected our results [48]. However, it has been reported that especially the lysosomal-dependent phagocytosis would actually be reduced and not increased in reactive astrocytes [49]”

  1. The authors showed the temporal changes in the amount of astrocyte-mediated synapse elimination in vivo and in OBSCs. However, how are the temporal changes in the amount of the whole (not only "astrocyte-mediated" but also other types of) synapse elimination (synaptophysin+/LAMP1+ spots) in vivo and in OBSCs? 
  • Thank you for this interesting suggestion. We tried to quantify overall changes in synaptophysin+/LAMP1+ spots. However, the background noise in our IHC stainings does not allow a consistent and reliable quantification of synapse elimination neither in vivo nor in OBSCs. Without an additional thorough evaluation of different cell types putatively involved in synapse elimination (e.g. microglia cells, oligodendrocyte progenitor cells, etc), we think that this quantification may not add clarity to the present work and the experiments necessary to investigate it at the single-cell level might go beyond the scope of this manuscript.

  1. Although the temporal changes in synaptophysin expression in vivo are shown in Figure 2, those in OBSCs are not shown in Figure 4. How are the temporal changes in synaptophysin expression in OBSCs? 
  • To address this concern, we have additionally quantified synaptophysin changes in OBSCs. All results about these quantifications have been pooled together in Supplementary Figure 2, as they provide additional insights but do not significantly contribute to the understanding of astrocyte-dependent changes in synapse elimination. As evident in this supplementary data, variations in synaptophysin levels among slices are observed, consistent with the inherent circuit rearrangements occurring in explants. Notably, despite these variations, astrocyte-dependent phagocytosis remains stable. This stability suggests that robust cell-autonomous effects may exert a governing influence on its activity.

  1. page 14, lines 479-481: The hypothesis of "developmental programmed reset after cutting" by the authors is very interesting but was not proved at all. Is there any evidence for this hypothesis? For example, if brain slices are cut at P0 or P7, do both slices show the same temporal changes in MEGF10 expression and phagocytic index?
  • Unfortunately, we cannot currently provide any experimental eveidence for our hypothesis. The suggested experiment might be useful, although evidences from other publications may limit its execution. It has been proved that late born neurons are still migrating to the upper layers until postnatal day (P)7. Any earlier tissue resection may affect neuronal development and the formation of synaptic contacts, which would in turn negatively impact our experimental results.

However, if this experiment is considered strictly necessary, we are willing to perform it with more time given for the revision.

Minor:

  1. It is better to explain about MEGF10 in a little more detail in Introduction. 
  • We have added two sentences in the introduction to shortly explain about MEGF10 (lines 59-64:”MEGF10, the mammalian ortholog of the glia-specific phagocytic proteins Draper in Dro-sophila and CED- 1 in C. elegans, ia a transmembrane receptor almost exclusively ex-pressed in astrocytes [15,16]. MEGF10 can mediate its phagocytic functions on apoptotic material via the recogni-tion of the “eat-me signal” C1q bound to phosphatidylserine (PS) exposed on dy-ing cells [17]”)

  1. Materials and Methods 2.2.2 and 2.3.5: The expression "tertiary antibody solution" is inadequate because this solution does not contain antibody. 

  In addition, this solution contains Avidin. However, I cannot find the result of staining with Avidin. 

  • Thank you for this observation. We have amended the erroneous description in lines 179-180 and 270-271 (“After washing, a solution containing DAPI (1:1000, Merck, Cat#32670) and Alexa Fluor 647-conjugated Streptavidin…)
  • We have also renamed the Avidin compound to a form which should be less misleading: we used a fluorophore-conjugated Streptavidin that binds with high affinity to the biotin-conjugated secondary anti-rabbit antibody to enhance detectability of low expressed targeted proteins.

  1. page 13, lines 468-469: Although the authors wrote "so far used to investigate hippocampal developmental processes", the use of OBSCs is not limited to the hippocampus. In particular, references of investigations with cerebellar OBSCs are lacking. 
  • Thank you very much for the observation, indeed we did not mention investigations with cerebellar OBSCs. We have now added some references to the use of cerebellar OBSCs in the (new) line 525.

Comments on the Quality of English Language

English can be polished throughout the manuscript. 

  • Following this suggestion, we have thoroughly revised the whole manuscript.

Round 2

Reviewer 1 Report

Comments and Suggestions for Authors

authors answered most of my concern. it is much better now.

Reviewer 2 Report

Comments and Suggestions for Authors

The authors have fully responded to my comments. I have no further comments.